

# Satellite tagging highlights the importance of productive Mozambican coastal waters to the ecology and conservation of whale sharks

Christoph A. Rohner[1], Anthony J. Richardson[2,3], Fabrice R. A. Jaine[1,4,5], Michael B. Bennett[6], Scarla J. Weeks[7], Geremy Cliff[8,9], David P. Robinson[10], Katie E. Reeve-Arnold[11] and Simon J. Pierce[1]

[1] Manta Ray & Whale Shark Research Centre, Marine Megafauna Foundation, Praia do Tofo, Mozambique
[2] CSIRO Oceans and Atmosphere, Dutton Park, QLD, Australia
[3] Centre for Applications in Natural Resource Mathematics (CARM), School of Mathematics and Physics, The University of Queensland, St Lucia, QLD, Australia
[4] Sydney Institute of Marine Science, Mosman, NSW, Australia
[5] Department of Biological Sciences, Macquarie University, North Ryde, NSW, Australia
[6] School of Biomedical Sciences, The University of Queensland, St Lucia, QLD, Australia
[7] Biophysical Oceanography Group, School of Geography, Planning and Environmental Management, The University of Queensland, St Lucia, QLD, Australia
[8] Kwa-Zulu Natal Sharks Board, Umhlanga, KZN, South Africa
[9] Biomedical Resource Unit, University of KwaZulu-Natal, Durban, KZN, South Africa
[10] Shark Watch Arabia, Dubai, United Arab Emirates
[11] All Out Africa Marine Research Centre, Praia do Tofo, Inhambane, Mozambique

Corresponding author
Christoph A. Rohner,
chris@marinemegafauna.org

## ABSTRACT

The whale shark *Rhincodon typus* is an endangered, highly migratory species with a wide, albeit patchy, distribution through tropical oceans. Ten aerial survey flights along the southern Mozambican coast, conducted between 2004–2008, documented a relatively high density of whale sharks along a 200 km stretch of the Inhambane Province, with a pronounced hotspot adjacent to Praia do Tofo. To examine the residency and movement of whale sharks in coastal areas around Praia do Tofo, where they may be more susceptible to gill net entanglement, we tagged 15 juveniles with SPOT5 satellite tags and tracked them for 2–88 days (mean = 27 days) as they dispersed from this area. Sharks travelled between 10 and 2,737 km (mean = 738 km) at a mean horizontal speed of $28 \pm 17.1$ SD km day$^{-1}$. While several individuals left shelf waters and travelled across international boundaries, most sharks stayed in Mozambican coastal waters over the tracking period. We tested for whale shark habitat preferences, using sea surface temperature, chlorophyll-*a* concentration and water depth as variables, by computing 100 random model tracks for each real shark based on their empirical movement characteristics. Whale sharks spent significantly more time in cooler, shallower water with higher chlorophyll-*a* concentrations than model sharks, suggesting that feeding in productive coastal waters is an important driver of their movements. To investigate what this coastal habitat choice means for their conservation in Mozambique, we mapped gill nets during two dedicated aerial surveys along the Inhambane coast and counted gill nets in 1,323 boat-based surveys near Praia do Tofo. Our results show that, while whale sharks are capable of long-distance oceanic movements, they can

spend a disproportionate amount of time in specific areas, such as along the southern Mozambique coast. The increasing use of drifting gill nets in this coastal hotspot for whale sharks is likely to be a threat to regional populations of this iconic species.

## INTRODUCTION

Knowledge of the movements of a species in space and time improves understanding of its habitat use and ecology, can enhance conservation management, and allows prediction of the species' response to changing conditions (*Sims, 2010*; *Block et al., 2011*; *Hays et al., 2016*). It can, however, be technologically and logistically challenging to study the movements of difficult-to-access species, such as wide-ranging marine fishes. Recent improvements in the equipment available for marine animal tracking, coupled with refined analytical techniques (*Nathan et al., 2008*; *Block et al., 2011*; *Costa, Breed & Robinson, 2012*), have made it easier to interpret both the movements and motivation underpinning the spatial ecology of even highly-mobile species (*Sims et al., 2006*).

Whale sharks *Rhincodon typus* move thousands of kilometres horizontally (*Hueter, Tyminski & De la Parra, 2013*; *Berumen et al., 2014*; *Hearn et al., 2016*) and perform vertical dives to >1,900 m depth (*Tyminski et al., 2015*). Although they actively move and do not simply follow surface ocean currents (*Sleeman et al., 2010*), ecological drivers of their movements are poorly understood. As coastal aggregations of whale sharks, including our study population off Mozambique, comprise mostly juveniles (*Rohner et al., 2015b*), reproduction is not likely to influence their movements during this life stage. Avoiding predation is also not a likely factor driving the movements of these large (>4 m in length) sharks that have few natural predators (*Rowat & Brooks, 2012*). Rather, prey search behaviour is likely to be the major driver of their movement, as zooplankton, the primary prey of whale sharks, are patchily distributed (*Lalli & Parsons, 1997*) throughout the species' tropical to warm temperate distribution (*Rowat & Brooks, 2012*).

Whale sharks are sighted off Praia do Tofo in southern Mozambique throughout the year (*Rohner et al., 2013b*; *Haskell et al., 2015*). Although some inter-annual site fidelity has been observed (*Rohner et al., 2015b*), photo-identification data suggest a short mean residency time (9 days) for this stretch of coast (C Prebble et al., 2017, unpublished data). Where they go, and the underlying drivers of this rapid turnover, remain uncertain. Although whale sharks are also seen in nearby Tanzania, Seychelles and Djibouti, photo-identification has shown limited connectivity among those sites (*Norman et al., 2017*; *Brooks et al., 2010*; *Andrzejaczek et al., 2016*). Despite their well-documented ability to move long distances (*Hueter, Tyminski & De la Parra, 2013*; *Hearn et al., 2016*), including from Praia do Tofo (*Brunnschweiler et al., 2009*), in the Indian Ocean there have been few examples of whale sharks being re-sighted outside the geographic region where they were first identified (*Norman et al., 2017*). As most photo-identification and tag deployment has taken place

at aggregation sites dominated by juvenile males, limited inference can be made about the behavior of the broader whale shark population (*Rohner et al., 2015b*). Mature whale sharks (>800–900 cm long; *Acuña Marrero et al., 2014*; *Rohner et al., 2015b*) may range further, and are likely to be more oceanic, as few have been sighted at coastal aggregation sites (*Hearn et al., 2016*; *Robinson et al., 2016*; *Ramírez-Macías et al., 2017*).

There is a clear conservation imperative to understand the movement ecology of whale sharks in southern Mozambique. Whale shark sightings at Praia do Tofo decreased by 79% between 2005 and 2011 with local environmental parameters taken into consideration (*Rohner et al., 2013b*), a trend that has continued following the conclusion of that study (*Pierce & Norman, 2016*). In the northern Mozambique Channel, following a slight increase in sightings from the tuna purse-seine fleet between 1991–2000, there was a decrease from 2000–2007 (*Sequeira et al., 2013*). In absolute terms, 600 sightings were reported from 1990s, decreasing to ∼200 from 2000–2007 (*Sequeira et al., 2014*), and peak monthly sightings decreased by ∼50% (*Sequeira et al., 2014*). While large-scale oceanographic mechanisms may influence sightings (*Rohner et al., 2013b*), there are also fisheries-related captures and mortalities of whale sharks in the region (*Jonahson & Harding, 2007*; *Capietto et al., 2014*; *Everett et al., 2015*).

Mozambique ranks low on the global Human Development Index: 0.418 = 181 of 188 countries (*United Nations Development Programme, 2016*). With over two thirds of Mozambique's population living within 150 km of the coast, ∼50% of their protein intake comes from fish (*Hara, Deru & Pitamber, 2007*). Gill net use has been increasing in Mozambique since the cessation of conflict in 1992 (*WWF Eastern African Marine Ecoregion, 2004*), and nets have been actively distributed by fisheries officials in some areas of the country to move fishing effort away from sensitive inshore nursery habitats (*Leeney, 2017*). Floating gill nets, extending from the beach to ∼200 m offshore, pose a threat to marine megafauna species swimming along this coast. While few formal data are available, these gill nets are routinely used off the Inhambane coast. At least two whale shark mortalities have been observed in this area, both sighted opportunistically (S Pierce, pers. obs., 2015), and entanglements are commonly reported (*Speed et al., 2008*; S Pierce, 2017, unpublished data). Whale sharks are a valuable focal species in marine tourism off Praia do Tofo and adjacent areas (*Pierce et al., 2010*; *Tibiriçá et al., 2011*; *Haskell et al., 2015*). The species received formal protection in Mozambique and, separately, were listed on Appendix I of the Convention of Migratory Species—which requires prohibition of take by signatory countries (which includes Mozambique)—during 2017.

Here we examine the regional movements and underlying environmental drivers of whale shark activity in Mozambique. We use aerial surveys, satellite telemetry and randomised model shark tracks to establish their activity hotspots in this region, and test the hypothesis that they preferentially spend most of their time in shallow coastal waters. With the limited data available, we also assess the potential for interaction with the coastal gill net fishery along the Inhambane coast.

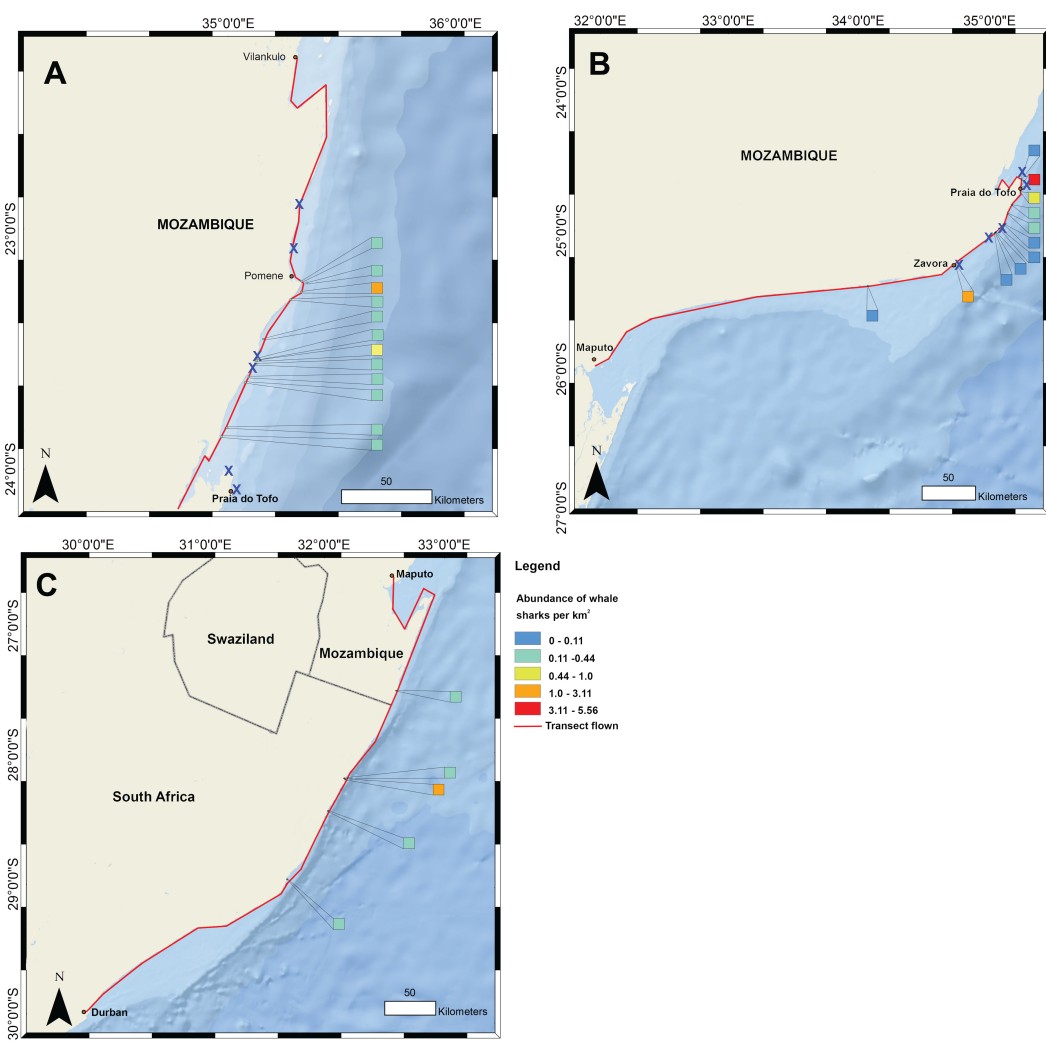

**Figure 1** **Whale shark and gill net locations from aerial surveys (conducted in 2004–2008 and in 2016, respectively).** Density of whale shark sightings along (A) the northern and (B) southern stretch of the southern Mozambique coast and (C) along the northern South Africa coast. The red line shows the flight path of whale shark surveys and a cross indicates gill nets in use.

# MATERIALS AND METHODS

## Aerial surveys for whale sharks

Data on the spatial distribution of whale sharks in southern Mozambique were acquired from aerial survey flights conducted by the KwaZulu-Natal Sharks Board in a top wing aircraft, flown 305 m (1,000 ft) above sea level at 184 km h$^{-1}$ (100 knots) (Fig. 1). Two observers recorded time and GPS coordinates for each whale shark within ~750 m of the coast during 10 regional flights between 2004 and 2008 in February and March. Flights were conducted when viewing conditions were optimal, characterised by light winds and minimal cloud (see full methods in (*Cliff et al., 2007*)). For aggregations of multiple individuals, central coordinates were used when only the start and end GPS position were

recorded. Aerial surveys have the limitations that whale sharks can only be seen by observers in surface waters, but the species also occupies deeper habitats in which they would not be able to be sighted. Logistical and cost constraints also meant that a relatively small number of aerial surveys were available for this study. Aerial survey data did not temporally match satellite tagging data. Spatial data were mapped in ArcGIS 10.2.1 in 1 km$^2$ grids and whale shark numbers expressed per km$^2$.

## Study area and whale shark tagging

Fifteen juvenile whale sharks, comprising 12 males and 3 females ranging from 540–865 cm total length (TL), were equipped with Smart Position or Temperature Transmitting (SPOT5) tags from Wildlife Computers, and tracked between November 2010 and January 2012. All tagged sharks were photographically identified based on their spot pattern posterior to the gills and matched on, or added to, the *Wildbook for Whale Sharks* global whale shark database (http://www.whaleshark.org; *Arzoumanian, Holmberg & Norman, 2005*). Sex was determined based on the presence (male) or absence (female) of claspers. Male maturity status was assigned according to clasper length and thickness (*Rohner et al., 2015b*). Longer-term (pre- and post-tagging) site fidelity of these sharks was assessed through to the end of 2016 via photo-identification submissions to the Wildbook database. Length estimates were derived from laser photogrammetry and visual size assessments, with an estimated error of ±50 cm (*Rohner et al., 2011*). All tags were deployed immediately off Praia do Tofo in southern Mozambique (23.85°S, 35.54°E). The tag's float was covered with dark antifouling paint to minimise bio-fouling and make it less obvious to predatory fishes. The tag was connected to a ∼5 cm titanium dart (Wildlife Computers) via a ∼180 cm tether. The first five tags had a stainless steel game-fishing swivel 30 cm from the dart, before it became evident from retrieval of shed tags that the swivel was a weak point and was therefore not used in later deployments. The first three tags used stainless steel wire as a short tether connecting the dart with the swivel; the remainder of the tether (and the entire tether in later deployments) comprised Dyneema braid. The dart was inserted into the skin at the posterior base of the 1$^{st}$ dorsal fin for the first three tags, using a 200 cm hand spear. Tag retention was improved on subsequent deployments by implanting the dart slightly further anteriorly, so that the tag floated adjacent to the 1$^{st}$ dorsal fin. No animal was restrained, caught or removed from its natural habitat for the purpose of this study. Whale shark tagging was compliant with ethics guidelines from the University of Queensland's Animal Ethics Committee and was conducted under their approval certificate GPEM/186/10/MMF/WCS/SF.

SPOT5 tags are positively buoyant and communicate with the ARGOS system (http://www.argos-system.org) when the wet/dry sensor is exposed to air. Tags were programmed for a daily limit of 300 transmissions to save battery power in case of extended tag retention. Transmitted data included tag location and accuracy (location classes 3, 2, 1, 0, A, B, Z), as well as sea surface temperature (SST) at the time of transmission. We used standard methods by *Hearn et al.* (*2013*; time of transmissions and time-at-temperature data) to determine when a tag detached from the shark, and removed the floating portion of the tracks before analyses were conducted. We only used location classes 3, 2 and 1

for further analyses. Estimated precision for location classes 3, 2 and 1 are theoretically 0.15, 0.35 and 1.00 km (ARGOS), but are larger when the tag is deployed on an animal at sea, with mean errors of 0.49, 0.94 and 1.10 km, respectively (*Costa et al., 2010*). More than half of all transmissions ($n = 1,930$) were characterised by ARGOS location classes 3, 2 and 1 and allowed accurate position estimation. Track distance was measured as the sum of the straight-line distances between two adjacent locations. Nine tags also recorded the proportion of time spent in 12 pre-defined temperature bins during 1, 5 or 6 h time intervals with data recorded at 05:00 h, 06:00 h, 11:00 h, 17:00 h, 18:00 h and 23:00 h. These time-at-temperature (TAT) data are limited to a period preceding a transmission via satellite, and hence do not reflect the full temperature range experienced by the tagged whale sharks. Available TAT data ranged from 36–100% of tracking days for individual sharks (mean = 81%) and 173 of 262 days in total for all sharks combined. SST and chlorophyll-*a* concentration (Chl-*a*) data were derived from the Moderate Resolution Imaging Spectroradiometer website (MODIS; modis.gsfc.nasa.gov) to produce monthly day- and night-merged SST and Chl-*a* time series at 1 km$^2$ spatial resolution for the period sharks were tagged. Chl-*a* was used as a proxy for zooplankton availability. Despite a possible lag in zooplankton abundance in response to a phytoplankton bloom (*Plourde & Runge, 1993*; *Flagg, Wirick & Smith, 1994*), phyto- and zooplankton abundance is often correlated (*Hutchinson, 1967*; *Richardson & Schoeman, 2004*; *Ware & Thomson, 2005*) and has been used similarly in previous studies on planktivorous elasmobranchs (*Sims et al., 2003*; *Sleeman et al., 2007*; *Graham et al., 2012*). To investigate drivers of coastal occurrences of whale sharks, SST values were extracted for one coastal location near Praia do Tofo (23.85°S, 35.62°E, 36 m depth) and one further offshore (23.85°S, 36.00°E, 988 m depth, ~45 km from the coast). SST and Chl-*a* values were also extracted for all positions with a location class 3, 2 or 1 from tracked whale sharks and for all positions from random model sharks (see below). A nine-month mean was produced for SST and Chl-*a*, encompassing all months when tagged sharks were tracked. Bathymetric data were derived from the NOAA ETOPO2 dataset at a ~1 km resolution.

## Random model sharks

We generated random model tracks ('model sharks') for each tagged shark ('real sharks') based on characteristics of the real tracks, similar to analyses conducted on basking sharks *Cetorhinus maximus* by *Sims et al. (2006)*. Input data for this analysis were observed locations with accuracy classes 3, 2 and 1, and a step was defined as the most direct, straight line between successive locations. Each model shark had the same starting location, overall track distance, and step-length frequencies as the real whale shark, but the order of steps was randomised. Real whale sharks often swam along the coast (Fig. S1), but as we had no *a priori* expectation whether sharks would move north or south or offshore, our random sharks took a random angle between steps while constraining the total length of the track to that of the real sharks. For a step that crossed land, or extended beyond the study area boundary (20–30°S, 31–40°E), another random turning angle was taken. The simulation was run in R (*R Development Core Team, 2008*) and sets of 100 model shark tracks were generated for each whale shark (Fig. S2). The aim of the model sharks was not to mimic

the real sharks, but to test whether the real sharks had a preference for locations on the regional shelf (0–200 m depth, 22.17°S–24.51°S), or for certain SST or chl-*a* conditions.

### Kernel density estimation analysis

All transmitted tag locations and modelled shark locations were input to ArcGIS 10.2.1. The "kernel density tool" was used to calculate percentile kernels of location density. Kernel density estimates were produced following *MacLeod (2013)*, with a search radius of 5 km and the outlying locations falling into the 2.5% kernel removed. Kernel density estimation analysis is based on transmitted locations and cannot consider the periods of the overall tracking duration when no locations were transmitted, which equaled 183 of 403 days in our dataset.

### Gill nets

Gill nets in the study area were set and drifting at the surface perpendicular to the beach. Net dimensions varied among fishing communities in the region, but were typically 20–200 m long, 5–8 m deep, and had a mesh size of 5–20 cm. Nets were made from monofilament or thin rope. Whale sharks are not specifically targeted in Mozambique, but nets with a larger mesh size present an entanglement risk. Locations of these gill nets along the ∼200 km of coastline between Zàvora to Pomene were recorded with a GPS during two aerial survey flights in May 2016. A transect was flown along the coast in a Bat Hawk LSA at 244 m (800 ft) above sea level at 60 knots and ∼300–500 m from the beach. To assess the trend in gill net use over time, we used survey data off the Praia do Tofo area itself. We conducted 1,323 boat-based surveys from 2012 to 2015, during which gill nets were counted on the way to dive sites located along a 40 km stretch of coast. Surveys were on average 21.3 km long, but survey design was influenced by which sites the dive company accessed at the time. We calculated the number of gill nets per 1,000 km of survey track for each year over the 4-year period. The gill net surveys did not temporally match with the whale shark tracking data, as pre-2012 gill nets were not counted because they were rarely in use around Praia do Tofo.

## RESULTS

### Whale shark aggregation

Flight observers recorded a total of 202 whale sharks in southern Mozambique during the 10 aerial survey transects between 2004 and 2008, with a mean of 3.4 individuals 100 km$^{-1}$. The focal area of whale shark sightings was the 200 km stretch of coastline between Zàvora and Pomene, with the peak at Praia do Tofo (Fig. 1). Several large aggregations were observed near Praia do Tofo, with the largest being 51 individuals sighted on 1 March 2005.

Gill nets were recorded during aerial surveys in the same region where whale shark sightings were highest between Zàvora and Pomene (Fig. 1). In the immediate area around Praia do Tofo, boat-based surveys showed that gill net usage increased ∼7 times from 0.95 to 6.44 nets per 1,000 km survey track from 2012 to 2015.

**Table 1 Track details of 15 whale sharks equipped with SPOT5 tags, with track number, shark ID on the Wildbook for Whale Sharks global database, sex, total length (TL), track start and end date and track duration.** Track distance is measured as the sum of the straight-line distances between two adjacent locations, only including locations of ARGOS class (LC) 3, 2 and 1.

| # | ID | Sex | TL (cm) | Start date | End date | Days | Track distance (km) | Speed (km day$^{-1}$) | No. of fixes (Pos. day$^{-1}$) | Number of fixes (LC 3,2,1 day$^{-1}$) | Days with locations (% of total tracking days) |
|---|---|---|---|---|---|---|---|---|---|---|---|
| 1 | MZ-421 | M | 560 | 11-Nov-10 | 14-Nov-10 | 4 | 66.6 | 16.7 | 8.7 | 6.7 | 4 (100%) |
| 2 | MZ-562 | M | 540 | 02-Feb-11 | 05-Feb-11 | 4 | 280.3 | 70.1 | 9.7 | 4.7 | 3 (75%) |
| 3 | MZ-286 | F | 550 | 19-Jul-11 | 28-Jul-11 | 10 | 261.5 | 26.1 | 6.9 | 4.2 | 8 (80%) |
| 4 | MZ-275 | M | 745 | 22-Jul-11 | 25-Jul-11 | 4 | 10.4 | 2.6 | 6.0 | 2.3 | 2 (50%) |
| 5 | MZ-418 | M | 700 | 09-Aug-11 | 18-Aug-11 | 10 | 325.5 | 32.6 | 7.1 | 2.6 | 10 (100%) |
| 6 | MZ-238 | M | 600 | 09-Aug-11 | 24-Aug-11 | 16 | 412.7 | 25.8 | 5.4 | 2.0 | 10 (63%) |
| 7 | MZ-241 | M | 630 | 10-Aug-11 | 03-Sep-11 | 25 | 814.6 | 32.6 | 5.4 | 2.9 | 23 (92%) |
| 8 | MZ-463 | M | 635 | 11-Aug-11 | 21-Aug-11 | 11 | 457.1 | 41.6 | 8.4 | 5.6 | 6 (55%) |
| 9 | MZ-606 | M | 550 | 26-Aug-11 | 20-Sep-11 | 26 | 668.0 | 25.7 | 7.8 | 3.8 | 21 (81%) |
| 10 | MZ-607 | M | 865 | 11-Aug-11 | 05-Oct-11 | 56 | 204.5 | 3.7 | 1.0 | 0.3 | 8 (14%) |
| 11 | MZ-600 | M | 600 | 23-Jul-11 | 18-Oct-11 | 88 | 2,446.8 | 27.8 | 5.1 | 3.2 | 38 (43%) |
| 12 | MZ-614 | M | 600 | 12-Oct-11 | 08-Nov-11 | 28 | 677.0 | 24.2 | 8.6 | 3.6 | 24 (86%) |
| 13 | MZ-615 | F | 650 | 26-Oct-11 | 17-Jan-12 | 84 | 2,736.7 | 32.6 | 3.7 | 1.6 | 38 (45) |
| 14 | MZ-165 | M | 670 | 25-Nov-11 | 26-Nov-11 | 2 | 23.9 | 11.9 | 12.0 | 6.0 | 2 (100%) |
| 15 | MZ-471 | M | 820 | 28-Nov-11 | 01-Jan-12 | 35 | 1,687.0 | 48.2 | 6.0 | 3.7 | 23 (66%) |
| Maximum | | | 865 | | | 88 | 2,737 | 70.1 | 12.0 | 6.7 | 100% |
| Minimum | | | 540 | | | 2 | 10 | 2.6 | 1.0 | 0.3 | 14% |
| Mean | | | 648 | | | 26.9 | 738 | 28.1 | 5.0 | 2.6 | 55% |

## Horizontal movements, tag retention and transmissions

SPOT5 tags remained on the sharks for 2–88 days (mean $\pm$ SD = 27 $\pm$ 28.1 d) and transmitted locations on 55% of days of the combined tracking duration (Table 1). Whale sharks travelled at a mean speed of 28 km day$^{-1}$ (median = 26.1 km day$^{-1}$, range = 2.6–70.1 km day$^{-1}$), similar to whale sharks tracked elsewhere (Table 2). The longest straight-line, along-track distances were 2,737 km over 84 days, and 2,447 km over 88 days (Table 1). All sharks remained within the southern Mozambique Channel and eastern South African waters while tagged (Fig. 2). Seven sharks (47%) moved offshore for at least part of their track, while the other eight (53%) remained on the shelf near the coast. Tracking duration did not influence whether sharks went offshore or stayed coastal ($t = -1.11$, $df = 11.4$, $p = 0.29$). Season may have played a role, with a greater proportion of sharks moving offshore in summer (three out of three), less in winter (three of five), and a lower proportion again in spring (two of seven), although numbers were too small to be conclusive (Fig. 2). Whale sharks travelling away from the coast swam significantly further (mean = 1,137 *vs.* 282 km) and faster (mean = 43 *vs.* 20 km day$^{-1}$) than those that stayed in coastal waters ($t = 2.29$, $df = 8.3$, $p = 0.05$, and $t = 2.46$, $df = 11.1$, $p = 0.031$, respectively). Of the five sharks tagged within a short time period (9–11 July 2011), one initially swam northward along the

**Table 2** Published whale shark tagging study information, with tag type; N, number of tracked sharks; M, males; F, females; mean total length and range in brackets (cm); mean (± SD) total distance travelled; tag attachment duration and mean (± SD) daily speed. Failed tags are not included in the analysis.

| Location | Tag type | N (M, F) | Total length (cm) | Distance (km) | Duration (days) | Speed (km d⁻¹) | Reference |
|---|---|---|---|---|---|---|---|
| Mozambique | Real-time | 15 (12, 3) | 648 (540–865) | 738 (±861.7) | 26 (±28.0) | 29 (±30.7) | This study |
| Qatar | Real-time | 28 (17, 11) | 704 (500–900) | 378 (±546.3) | 69 (±60.7) | 7 (±13.5) | *Robinson et al. (2017)* |
| Ecuador | Mix | 26 (0, 26) | 1047 (400–1,310) | 2,273 (±1,933.6) | 62 (±50.6) | 41 (±25.5) | *Hearn et al. (2016)* |
| Saudi Arabia | Archival | 47 (14, 16) | 391 (300–700) | 502 (±613.4) | 146 (±80.3) | 4 (±4.9) | *Berumen et al. (2014)* |
| Mexico | Archival | 28 (10, 18) | 738 (500–900) | 699 (±1,322.8) | 68.4 (±54.5) | 9 (±11.0) | *Hueter, Tyminski & De la Parra (2013)* |
| Mozambique | Archival | 2 (1, 1) | 725 (650–800) | 607 (±838.6)* | 47 (±56.6) | 8 (±8.3) | *Brunnschweiler et al. (2009)* |
| Seychelles | Real-time | 3 (1, −) | 617 (500–700) | 1,769 (±1,471.2) | 42 (±20.8) | 43 (±70.6) | *Rowat & Gore (2007)* |
| Taiwan | Real-time | 3 (3, 0) | 423 (400–450) | 4,250 (±1,458.1) | 143 (±56.1) | 30 (±26.0) | *Hsu et al. (2007)* |
| Australia | Archival | 10 (1, 7) | 715 (470–1,100) | 581 (±544.8)* | 92 (±88.9) | 6 (±6.1) | *Wilson et al. (2006)* |
| SE Asia | Real-time | 6 (−, −) | 567 (300–700) | 890 (±1,284.1) | 35 (±48.9) | 25 (±26.2) | *Eckert et al. (2002)* |
| Mexico | Real-time | 14 (−, 7) | 643 (300–1,800) | 1,812 (±3,749.4) | 149 (±334.6) | 12 (±11.2) | *Eckert & Stewart (2001)*** |

**Notes.**

*Indicates straight-line distances from tagging to pop-up location.

**A record of a >13,000 km track from this paper is now broadly considered to be from a floating tag (*Andrzejaczek et al., 2016*).

coast and four swam southward. Apart from MZ-463, which travelled to northern South Africa, these sharks stayed in coastal waters and swam past Praia do Tofo again after 3–13 days.

## Home range and random model sharks

The kernel density estimation analysis of whale shark tracks showed that the main hotspot of whale shark activity was between Zàvora and Praia do Tofo, with a second, less intense hotspot around the Pomene headland, 100 km north of Praia do Tofo (Fig. 3A). High-use areas were on the continental shelf. By contrast, model sharks spread from Praia do Tofo and their high activity zone included areas off the continental shelf (Fig. 3B). Overall, whale sharks spent significantly more time on the regional shelf (85%) than model sharks (15%; $\chi^2 = 1239.6$, $df = 15$, $p < 0.001$). An example is shark MZ-241, which swam north along the coast, then briefly headed offshore, before returning to coastal waters south of Praia do Tofo (Fig. S2). This was one of 10 sharks that spent more time on the shelf than any of the corresponding 100 model tracks for each real shark. Only MZ-562 (8% of a 3-day track) and MZ-463 (26% of a 10-day track) spent less time on the regional shelf than half of the model sharks.

Tagged sharks transmitted their position on 30 separate days while they were in the immediate whale shark search area off Tofo (23.85°S–23.93°S), excluding detections from the day of tag deployment. Only two sharks, on two separate days, were re-sighted in regular visual surveys using photo-identification during the period of tag deployment. One of these had its tag entangled in a fishing line, causing the tag to sit under the shark's body and preventing it from breaking the surface to transmit, so we removed the tag and line. Photo-identification data indicated that most of the tagged sharks (67%) returned to the
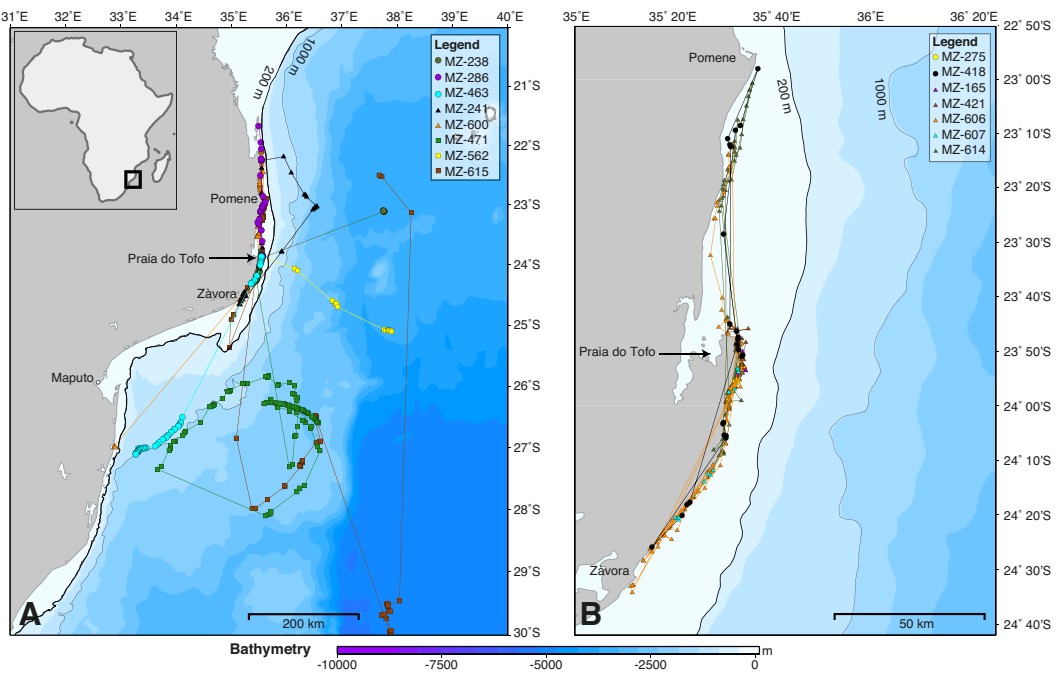

**Figure 2  Whale shark tracks in the southern Mozambique Channel.** Bathymetry maps showing the movements of satellite-tagged sharks. (A) Sharks that included large-scale movement off the continental shelf ($n = 8$). (B) All sharks that remained locally on the continental shelf ($n = 7$). Circle, winter; triangle, spring; square, summer deployments.

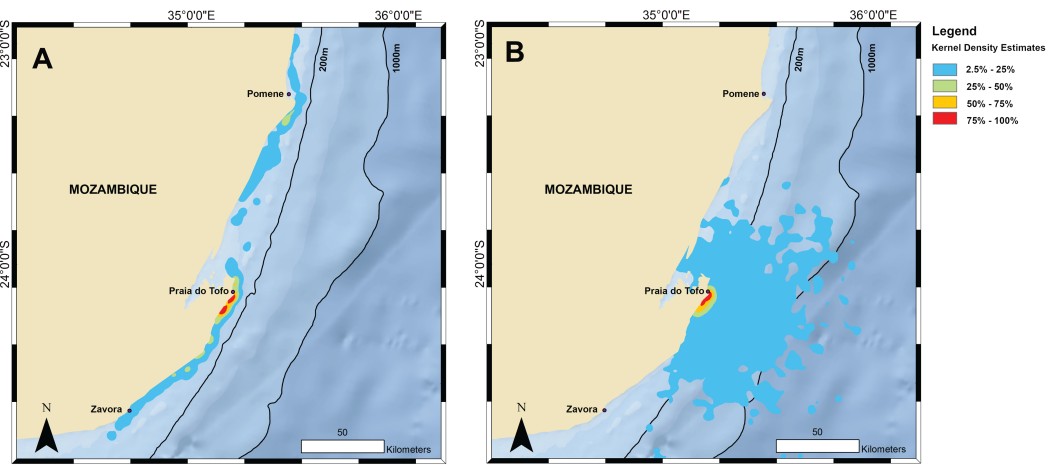

**Figure 3  Kernel density maps.** Kernel density estimations from all satellite tag locations for (A) tracked whale sharks and (B) random model sharks.

___

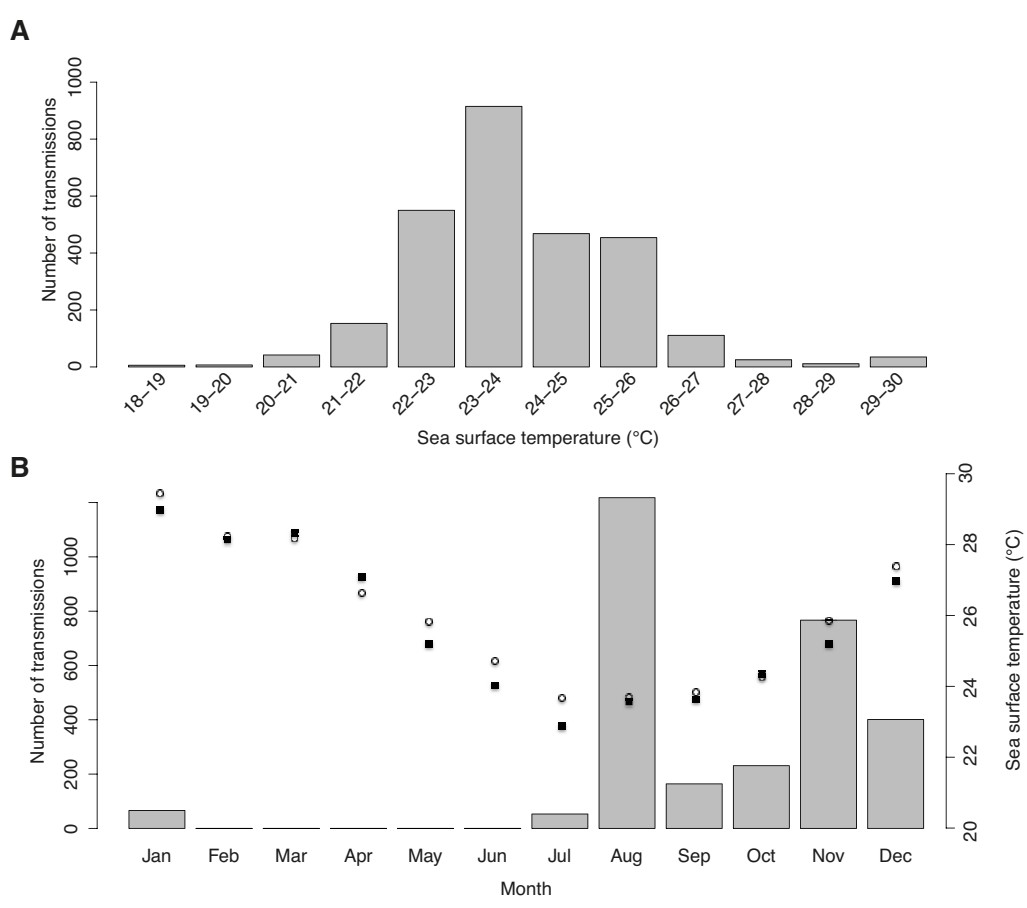

**Figure 4 Sea surface temperature preferences.** (A) Number of tag transmissions in each sea surface temperature bin, showing a wide temperature distribution and an affinity for surface temperatures of 22–26 °C. (B) Number of transmissions made by the tags in each month, with mean monthly sea surface temperature plotted for Praia do Tofo (square; 23.85°S, 35.62°E) and 45 km directly offshore (circle; 23.85°S, 36.00°E).

region after losing their tag, with these sharks being sighted on 2–11 unique days (mean = 4.8 ± 2.6 days) over 1–6 unique calendar years between 2005 and 2016 (mean = 3.2 ± 1.4 years).

## Temperature and chlorophyll-a distributions

Tag-derived temperature data showed whale sharks moved through surface temperatures between 18.5–29.7 °C, with a mean of 23.9 ± 1.51 °C. Half of all transmissions were from a narrow range of 22–24 °C waters, and >95% were from 21–27 °C waters (Fig. 4A). This temperature distribution is at least partly a result of the seasonal bias in tagging, with most transmissions in winter and spring when coastal and offshore temperatures were relatively cool (Fig. 4B).

Whale sharks spent more time in cooler water with higher Chl-$a$ than model sharks (Figs. 5A and 5B). Mean Chl-$a$ was significantly higher for whale sharks (mean = 1.18 ± 2.74 mg m$^{-3}$) than model sharks (mean = 0.27 ± 0.79 mg m$^{-3}$; $t = -9.38$, $df = 803.3$,

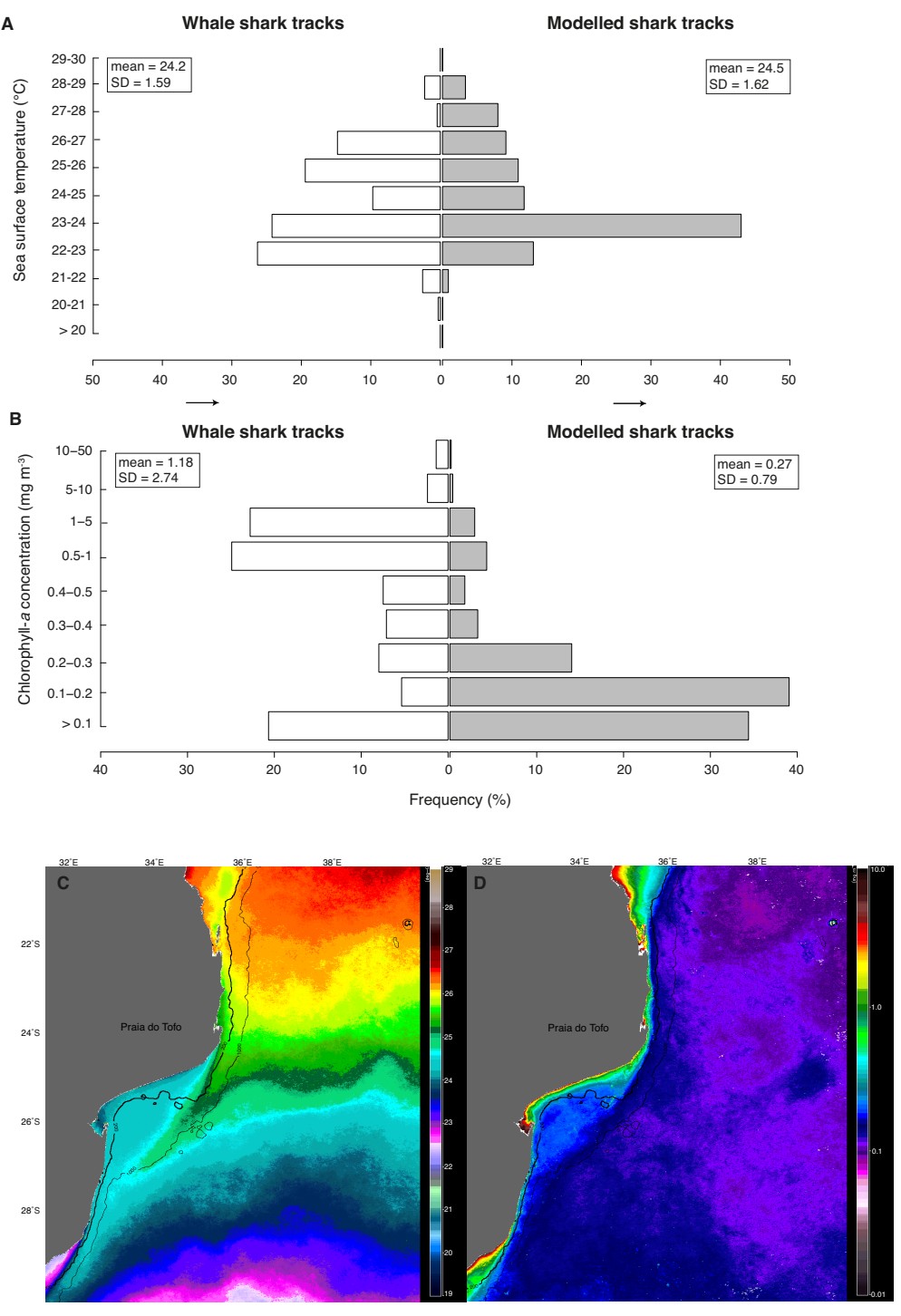

**Figure 5  Real vs. random tracks.** Distributions for all locations of real tracks ("whale shark tracks", white) and for all locations of 100 random tracks per real shark ("modelled shark tracks", grey) of satellite-derived (A) sea surface temperature (SST) and (B) chlorophyll-*a* concentration (Chl-*a*). Nine-month mean images of (C) SST and (D) Chl-*a* showing their respective mean regional distributions for the study period.

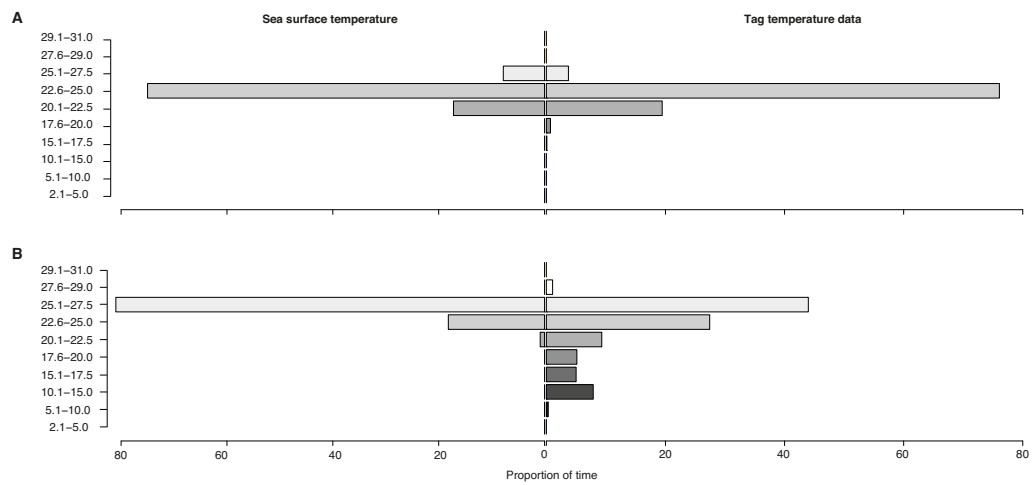

**Figure 6** **Sea surface vs. vertically-integrated temperatures.** Proportion of time spent in each temperature bin for sea surface temperature of all locations ("Sea surface temperature") and for tag-recorded, time-integrated temperature ("Tag temperature data") for locations (A) on the shelf and (B) off the shelf for all tags.

$p < 0.001$). Mean satellite-derived SST was significantly cooler for whale shark locations (mean $= 24.23 \pm 1.59$ °C) than for model sharks ($24.49 \pm 1.62$ °C; $t = 4.28$, $df = 679.4$, $p < 0.001$; Fig. 5B). Chl-*a* and SST distributions were also significantly different between whale sharks and model sharks ($\chi^2 = 549.1$, $df = 8$, <0.0001 and $\chi^2 = 297.5$, $df = 10$, $p < 0.0001$, respectively). Coastal shelf waters had higher Chl-*a* (Fig. 5C) and were cooler (Fig. 5D) than offshore waters over the 9-month duration of this study.

## Vertical movement (inferred from temperature-at-depth)
Temperatures recorded in binned intervals of up to 24 h prior to each transmission indicated that some of the tagged sharks made pronounced vertical movements. Combining data from all tags, the temperature bin extremes ranged from 5.1–10 °C up to 27.6–29 °C. The largest proportion of time (64%) was spent in 22.6–25 °C water. Overall, whale sharks experienced a wider temperature range when they were off the continental shelf as opposed to inshore (Fig. 6). When on the shelf, they spent the majority of time (76%) in 22.6–25 °C water, while the coldest temperatures recorded from shelf waters were in the 15.1–17.5 °C bin (0.1% of time). By contrast, when off the shelf, sharks spent the most time in warmer 25.1–27.5 °C water, while the coldest offshore temperatures were in the 5.1–10.0 °C (0.3% of time) and in the 10.1–15.0 °C bins (7.9%).

## DISCUSSION
Whale sharks tagged at Praia do Tofo moved widely in southern Mozambican and eastern South African waters. Although the duration of tag transmission was relatively short for most sharks, they spent a disproportionately high amount of time in regional shelf waters between Zàvora and Pomene. This is of concern for regional whale shark conservation, as gill net use is rapidly increasing in the same coastal area where tagged whale sharks spent

a lot of time, leading to a higher chance of net entanglement and mortality. Whale sharks moved through water with higher Chl-$a$ than simulated model sharks, suggesting that foraging is a major driver of their movements in this region.

## The coastal whale shark hotspot in southern Mozambique

The primary activity hotspot for tagged whale sharks was a ∼200 km stretch of shelf waters along the coast from Zàvora to Praia do Tofo, and also around Pomene. This agrees with our aerial survey data from 2004–2008, despite the temporal mismatch of the two datasets, which strengthens the importance of this area for whale sharks. One caveat is that both technologies require the sharks to be in surface waters to be detected, and whale sharks may also be abundant elsewhere in deeper water but remain undetected. The observed hotspot was not the result of random movement, or a bias due to the tagging site, as model sharks spent significantly less time on the continental shelf than real whale sharks. In addition, long-term whale shark sightings at Praia do Tofo fluctuated, but did not have a seasonal trend (*Rohner et al., 2013b*). Hence, while our tracks were relatively short and did not span the whole year, the general pattern may apply throughout the year. The narrow shelf waters around Praia do Tofo were a preferred habitat for whale sharks in the region in our study, which is further corroborated by photo-identification and tourism studies (*Pierce et al., 2010*; *Haskell et al., 2015*; *Rohner et al., 2015b*). However, our tagging data also show that the core use area for whale sharks in Mozambique is larger than previously reported, and larger than in some other, more defined whale shark aggregations that exploit specific and localised ephemeral prey sources or biological events (*Heyman et al., 2001*; *Robinson et al., 2013*; *Rohner et al., 2015a*). For example, the 50% kernel densities covered 185 km$^2$ in Mozambique compared to just 66 km$^2$ in Qatar (*Robinson et al., 2017*).

Eight whale sharks (53% of those tagged) returned to the tagging site during tag attachment after significant initial (>50 km) movement away from the site, mostly along the coast. Only two of these individuals were photographically recaptured, despite close to daily survey effort in good conditions for potential resightings (S Pierce, 2012, unpublished data). This further stresses the importance of sightings-independent methods for assessing whale shark residency, as detectability can be low, even when regular visual surveys are performed (*Cagua et al., 2015*; *Andrzejaczek et al., 2016*). Eight of the 15 tagged whale sharks were photographically re-sighted at Praia do Tofo after losing their tags, indicating some degree of site fidelity. Elsewhere, whale sharks also return to other aggregation sites, as determined by photo-ID techniques (*Holmberg, Norman & Arzoumanian, 2009*; *Rowat et al., 2011*), and their site fidelity may be more prevalent than expected from sightings data (*Cagua et al., 2015*).

## Preference for shelf waters

During the 8 months of the year (Jul–Feb) that whale sharks were tracked, over a combined duration of 403 days, whale sharks actively chose continental shelf waters that were cooler and had higher Chl-$a$ than the modelled sharks that moved randomly. While shallower, cooler water and higher Chl-$a$ co-vary in our study region, the bigger difference in Chl-$a$ between real and model sharks indicated that they mostly selected Chl-$a$. Their preference

for cooler shelf waters with higher Chl-*a* is thus likely to be related to foraging activities. Even though whale sharks do not directly feed on phytoplankton, and there is often a lag between the timing of phytoplankton and zooplankton blooms (*Plourde & Runge, 1993*; *Flagg, Wirick & Smith, 1994*), high phytoplankton biomass is often indicative of high zooplankton densities (*Hutchinson, 1967*; *Richardson & Schoeman, 2004*; *Ware & Thomson, 2005*). Whale shark sightings (*Sleeman et al., 2007*) and the abundance of other large marine animals have previously been correlated with Chl-*a* (*Zagaglia, Lorenzzetti & Stech, 2004*; *Block et al., 2011*; *Graham et al., 2012*; *Jaine et al., 2012*). We suggest that the juvenile whale sharks at Praia do Tofo that stay on the shelf do so to take advantage of high local food availability. Whale sharks off Praia do Tofo have been seen feeding ~20% of their time during daylight hours (*Pierce et al., 2010*). Stomach contents of whale sharks from southern Mozambique and northern South Africa were dominated by mysids, a group of demersal zooplankton that emerge into surface waters at night (*Rohner et al., 2013a*). Shallow coastal waters also have a high abundance of other demersal zooplankton (*Alldredge & King, 1977*; *Ohlhorst, 1982*). This suggests that Mozambican coastal waters are important foraging grounds for these juvenile whale sharks, perhaps more at night than during the day.

Tag-recorded temperature data further support the hypothesis that whale sharks often remain in shelf waters to exploit foraging opportunities. When off the shelf, in deeper waters, whale sharks experienced a broader temperature range that extended to cooler temperatures than those recorded from the surface. By contrast, the temperature range recorded for locations on the shelf were similar to surface water temperatures. This indicated that little diving behaviour took place, as shelf waters in the Mozambique Channel get significantly cooler at depth (*Lamont et al., 2010*; *Malauene et al., 2014*; *Rohner et al., 2017*). This suggested that whale sharks increased their vertical movement when off the shelf. Whale sharks dive to bathypelagic depths (>1,000 m), as has been demonstrated with pressure-recording tags (*Brunnschweiler et al., 2009*; *Tyminski et al., 2015*). One whale shark tagged near Praia do Tofo undertook most deep dives in the southern Mozambique Channel during the day, when zooplankton is often found at depth (*Loose & Dawidowicz, 1994*), suggesting that these dives might have been related to foraging (*Brunnschweiler et al., 2009*). Results from biochemical dietary studies have suggested that whale sharks may feed on meso- and bathypelagic crustaceans and fishes, among other prey (*Rohner et al., 2013a*). Since temperatures of 4.2 °C, 5.5 °C and 9.2 °C were recorded at 1,264 m, 1,092 m and 1,087 m depth respectively (*Brunnschweiler et al., 2009*), one of our tagged sharks, MZ-463, may have dived to depths of around 1,000 m (5.1–10 °C bin), potentially to feed.

Whale sharks swam at a mean speed of ~28 km d$^{-1}$ which is within the large range of swimming speeds reported in previous studies. Larger sharks (>900 cm TL) tagged in other locations exhibited similar speeds to juveniles (*Wilson et al., 2006*; *Hearn et al., 2016*), and the difference in distance covered per day among studies is likely to be primarily influenced by the sharks' behaviour (feeding vs. migrating) rather than their size, at least for sharks >400 cm TL. Similarly, total mean track distance in different studies is likely to be influenced by both tracking duration and whale shark behaviour.

## Conservation and management implications

This study supports the results from other tracking studies that show whale sharks routinely swim long distances and cross international boundaries. Offshore areas were used by some of the tagged individuals and may be important habitats for the species, particularly large, mature animals (*Hearn et al., 2016*) that are seldom seen at coastal aggregations (*Rowat & Brooks, 2012*; *Rohner et al., 2015b*; *Ramírez-Macías et al., 2017*). Results of this study indicate that southern Mozambican whale sharks routinely cross into South African waters, in addition to some interchange with Madagascar (*Brunnschweiler et al., 2009*), the Seychelles (*Andrzejaczek et al., 2016*) and Tanzania (*Norman et al., 2017*). A coordinated regional approach to managing the species' conservation in the Western Indian Ocean is therefore of importance, given the transnational boundaries crossed by individual sharks, and their occupancy of international waters.

That notwithstanding, these juvenile whale sharks spent a large proportion of their time on the shelf adjacent to Praia do Tofo, indicating that this is a particularly important habitat within the region. Drifting gill nets are set in the same areas where the whale shark activity hotspot was recorded. Furthermore, their use in the Praia do Tofo area has increased over recent years. While the satellite tracking dataset (2010–2012) does not temporally match with the gill net abundance dataset (2012–2015), we suggest that the spatial overlap of the whale shark hotspot and the increasing gill net use in the area raises concerns, especially considering the regular north-south movement of whale sharks close to the coast that is likely to bring them in contact with these nets. However, concomitant data on gill net numbers and locations and the distribution of whale sharks would be needed to quantify the risk to whale sharks. Other threatened species, such as manta rays, may also be affected by this fishery (*Rohner et al., 2017*). There are few available data on catch and injury rates along this remote coast, although multiple mortalities from gill nets and injuries characteristic of net entanglement have been reported from the Inhambane Province (*Speed et al., 2008*, S Pierce, 2015, unpublished data). Interview-based surveys with fishing communities are presently underway to provide more information on catches. Whale sharks within the Indian Ocean are listed as 'Endangered' on the IUCN Red List of Threatened Species (*Pierce & Norman, 2016*), and they are locally important to a burgeoning marine tourism industry (*Pierce et al., 2010*; *Tibiriçá et al., 2011*; *Haskell et al., 2015*). The lack of habitat-level protection, coupled with poor regulation of inshore fisheries in Mozambique, is a clear threat to this population.

## ACKNOWLEDGEMENTS

We thank Clare Prebble and Peter Bassett, along with other volunteers from the Marine Megafauna Foundation (MMF) for their assistance in the field. We thank the people who found and returned some of the tags. We gratefully acknowledge the NASA Ocean Biology Processing Group for provision of Moderate Resolution Imaging Spectroradiometer satellite data. Janneman Conradie and Joshua Axford from MMF conducted the gill net aerial surveys and Ross Newbigging (All Out Africa) and Jessica Williams (Moz Turtles) helped compile the gill net visual survey data. We thank David Johnston,

Jeremy Kiszka and one anonymous reviewer for their constructive comments on our submitted manuscript. Casa Barry Lodge and Peri-Peri Divers provided logistics field support. This research has made use of data and software tools provided by Wildbook for Whale Sharks, an online mark-recapture database operated by the non-profit scientific organisation Wild Me with support from public donations and the Qatar Whale Shark Research Project. Some maps were created using ArcGIS software by Esri, please visit http://www.esri.com. We acknowledge the use of free vector and raster map data sourced from http://www.naturalearthdata.com.

### Funding

Field work was supported by the Shark Foundation, Aqua-Firma, Waterlust, a Rufford Small Grant and the PADI Foundation. Christoph Rohner and Simon Pierce were supported by two private trusts. Anthony Richardson was supported by the Australian Research Council Future Fellowship FT0991722. The funders had no role in study design, data collection and analysis, decision to publish, or preparation of the manuscript.

### Grant Disclosures

The following grant information was disclosed by the authors:
Shark Foundation.
Aqua-Firma.
Waterlust.
PADI Foundation.
Australian Research Council Future Fellowship: FT0991722.

### Competing Interests

The authors declare there are no competing interests.

### Author Contributions

- Christoph A. Rohner conceived and designed the experiments, performed the experiments, analyzed the data, wrote the paper, prepared figures and/or tables, reviewed drafts of the paper.
- Anthony J. Richardson conceived and designed the experiments, analyzed the data, contributed reagents/materials/analysis tools, prepared figures and/or tables, reviewed drafts of the paper.
- Fabrice R.A Jaine analyzed the data, reviewed drafts of the paper.
- Michael B. Bennett and Geremy Cliff contributed reagents/materials/analysis tools, reviewed drafts of the paper.
- Scarla J. Weeks conceived and designed the experiments, contributed reagents/materials/analysis tools, reviewed drafts of the paper.
- David P. Robinson analyzed the data, prepared figures and/or tables, reviewed drafts of the paper.

- Katie E. Reeve-Arnold performed the experiments, reviewed drafts of the paper.
- Simon J. Pierce conceived and designed the experiments, performed the experiments, wrote the paper, reviewed drafts of the paper.

## Animal Ethics

The following information was supplied relating to ethical approvals (i.e., approving body and any reference numbers):

Whale shark tagging was compliant with ethics guidelines from the University of Queensland's Animal Ethics Committee and was conducted under their approval certificate GPEM/186/10/MMF/ WCS/SF.

## Data Availability

The raw data has been provided as a Supplemental File.

## Supplemental Information

Supplemental information for this article can be found online at http://dx.doi.org/10.7717/peerj.4161#supplemental-information.

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
