# Peer review of "Satellite tagging highlights the importance of productive Mozambican coastal waters to the ecology and conservation of whale sharks"

_PeerJ, doi:10.7717/peerj.4161_

## Round 0.1 · original submission · Major Revisions

Both reviewers provide detailed comments on how to address the remaining issues with the present manuscript. The authors are urged to address all of the points presented in both reviews. In particular, there needs to be more detail in the methods, and the short-term nature of tag deployments needs to be better addressed (e.g. addressing limitations of limited duration data and acknowledging issues around gaps in movement data - see below).

·

Basic reporting

The paper is clearly written and concise. Background information provided are relevant, the paper is relatively well structured, except in the abstract. Specific comments are provided below.

I did not identify issues with the figures, tables and raw data. However, I made some comments (general comments) for Fig. 1 that remains difficult to read. Results and hypotheses are relevant.

Experimental design

I have no comments here, except for the description of the methods that are lacking, particularly for aerial surveys.

There is a temporal mismatch between aerial survey and satellite tagging data. Since artisanal fisheries are dynamic in time and space, and that prey availability (influenced by the complex oceanography of the Mozambique Channel) also greatly varies between years, you need to clearly show how comparable both datasets are.

I know the authors cite Cliff et al. (2007) for details on the sampling for aerial survey data. However, more (even brief) information would be needed to enlighten the reader. You also need to refer to a map with the sampling design.

Validity of the findings

I have reservations on the use of gillnet data, and about the compatibility of aerial survey and satellite tag data. More details are provided in the "general comments" section. Movement data are robust and statistically sound, and conclusions are well stated. However, conclusions on the overlap between gillnets and whale sharks are speculative and should be identified as such.

Additional comments

Abstract

The abstract lacks of structure. The first sentence should be removed, and I would rather include one more sentence on aerial survey methods. A justification of why you combined aerial survey and satellite (even very brief) tagging methods would be good to add too. A better presentation of the question(s)/hypothesis is critically needed here too.


Introduction

L. 59-61: this sentence (last of 1st §) seems to be out of place. Should be removed.

L. 66-67: “ecological drivers of their movements is poorly understood” might be better.

L. 67-71: very general statement here, which also seems to be incomplete (how about predation risk?). General statements like this one should be in 1st §. 2nd § should keep focusing on whale sharks.

L. 81: “in review” paper should be removed (replaced by unpublished data), same for Normal et al. (l. 84, 89)

L. 96-106: anthropogenic threats might explain the observed decline. However, oceanographic conditions and changes in productivity (and prey availability) at feeding aggregations might have affected whale shark abundance. It should not be ignored here, and I think anthropogenic threats are obvious in this region (particularly bycatch in drift gillnets), but other drivers in abundance should be mentioned, even briefly.

L. 123: and “underlying environmental drivers” would be (a bit) better.


Materials and methods

There is a temporal mismatch between aerial survey and satellite tagging data. Since artisanal fisheries are dynamic in time and space, and that prey availability (influenced by the complex oceanography of the Mozambique Channel) also greatly varies between years, you need to clearly show how comparable both datasets are.

L. 140-141: I know you cite Cliff et al. (2007) for details on the sampling. However, more (even brief) information would be needed to enlighten the reader. You also need to refer to a map with the sampling design.

L. 227-228: what are net dimensions and where are they set? What is the mesh size? What kind of gillnets are used? It looks like these are bottom set since their location is very coastal. Unless you prove otherwise, I doubt whale sharks can get entangled in these, whereas large meshed drift gillnets can.

L. 232: “boat-based” surveys? Looks like these net locations were recorded opportunistically. More information on sampling is needed here. How representative this sampling is? Not sure if the sampling really reflects the distribution of gillnets in the study area.

Results

Fig. 1: very few gillnets were recorded, so I wonder how relevant it is to link their distribution and whale shark distribution and movements. You also mention in the methods that gillnets were recorded during “boat-based” surveys, but it is not clear whether gillnets on the map were recorded during flights or at-sea surveys.

L. 286: How did you find the animal?

L. 295: I think “range” would be better than “band”.

L. 366-367: sounds speculative. Any reference supporting this? What about thermoregulation?

L. 396-397: speculative. This statement needs to be.


Discussion

L. 325-332: it doesn’t look like you are really discussion the spatial overlap between gillnets and whale sharks. It really convinces me that this dataset might not be relevant for the scope of the paper.

L. 347: “in revision” papers should not be cited, same for submitted (l. 422).

L. 433: analyses that were performed do not clearly show the overlap between gillnet fisheries and whale sharks.

Reviewer 2 ·

Basic reporting

the manuscript is well written and easy to read. I'm uncertain if PeerJ allows "in revision" articles to be cited.

Experimental design

The limited aerial survey effort combined with diving behavior limits the utility of these surveys.

The short track duration and resulting data set on movement of tagged whale sharks limits the ability of this research to determine anything other than short term movement and habitat use. While the data are presented well and used to interpret selection of inshore habitats there is limited discussion on long term habitat use combining satellite and visual observations. By addressing the limitations of the short term data and acknowledging some of the issues around gaps in movement data for ~ half the year I think the manuscript would be greatly improved.

Methods of random models was not adequately addressed and I could not view figure 2 in the supplementary material. More detail on how the models were generated including the code used to generate these models would be useful.

Validity of the findings

See previous comments on limitations of aerial survey data as well as limitations of whale shark movement data due to short track duration and no data on movements for ~ 5 months of the year.
More could be done to determine level of overlap with gillnets sets along the coast.
The analysis would benefit from a seasonal component to movement analysis - did sharks occupy different areas at different times. The horizontal movements are very descriptive and while data are limited, more should be done to try and determine any patterns related to season or environmental conditions.

Additional comments

Introduction
Line 70 – there are more relevant papers on reproductive movements than the one cited. Please provide additional examples.

Line 84 – does the journal allow manuscripts in revision to be cited (Norman et al. in revision)?

Line 137 – 139: the amount of aerial survey effort is very limited and amounts to ~2 flights per year over 5 years. Furthermore the survey area is incredibly narrow. It is difficult to see how such a survey could provide data relevant on abundance given the variability in whale shark ‘availability’ to aerial surveys. It is well known that Whale Sharks spend considerable time in deep water (even when close to shore) making them impossible to see from the air. The variation in the number of sharks visible during any one flight makes it difficult to reconcile abundance estimates.

Line 243 – the large number of sharks seen on one day highlights the variability in whale shark abundance/sightability which do not always reflect true population size.

Line 291 – 297: that temperature data obtained from SPOT tags is limited to days when temperature histogram data were recorded and while there is a nice range of data presented, the authors need to acknowledge that these data are not reflective of the total range of temperature experienced by the sharks. The number of time at temperature messages received per day should be presented in a table to illustrate over what time period these data were recorded.

Figure 2: it is unclear from this figure whether the duration of the tracks influenced offshore movement. It would be good to colour code days since tagging to illustrate whether sharks that moved off the shelf were tracked for longer or whether this just represents individual variation. Most offshore movement seem to have occurred in summer suggesting some seasonal shift.


KUD estimates from satellite telemetry of whale sharks is problematic given low detection rates combined with periods of limited or no detections over multiple days. It is therefore important to present detection data in terms of number of fixes per day to enable the reader to interpret KUD estimates which will be biased by periods when the sharks were detected more frequently either due to favourable environmental conditions or because the sharks were spending more time at the surface swimming at slower speed resulting in more fixes.

Figure 2. the end of the track of shark MZ471 appears to be from the tag floating on the surface.

Figure 4 is misleading as the total number of transmissions will be a function of how many tags were deployed as well as weather conditions. The number of transmissions should therefore be an average per tag not the sum of all tags.

Figure 5: it would be good to see a figure of the randomised tracks to determine how realistic/unrealistic they are compared to recorded track.

Line 309-321: the fact that tags didn’t record depth limits the use of these data and this section appears to be overly long. I suggest shortening this section to focus on the main point - It is clear from the temperature data that sharks that moved off the shelf experienced a wider range of temperatures most likely due to vertical movements into cooler waters. It would be nice to try and match these temperatures with known depths if there are any oceanographic data available for this region.

Line 334-347: the short duration of the tracks and the fact that combined tracks do not span more than 7 months is a major limitation of the research. This limits the ability to make overarching comparisons and management implications need to be taken into context. While it is clear that sharks spend a disproportionate amount of time close to shore for some part of the year and return to the region over multiple years there is still uncertainty over where the sharks go and more needs to be done in the discussion to piece together this research combined with previous work.

Line 363-364: the limited time animals were tracked for needs to be acknowledged here. This statement should include the period animals were tracked.

Line 387-390: the lack of temperature stratification in relatively shallow coastal waters limits the ability to determine diving behaviour from temperature alone. The statement that little diving took place is therefore not supported by the data as there isn’t any data on diving behaviour in shallow coastal waters. It should also be acknowledged that data on temperature is not collected frequently by SPOT tags (how often temperature data were obtained is not presented so I’m assuming that the transmission of these data was limited as is often the case when prioritising location transmission over temperature data).

Line 401 – 404: given that there is very limited data on vertical movement that is inferred from temperature data it seems a long bow to claim that these data support a hypothesis that vertical movements in whale sharks relate to foraging behaviour. I suggest removing this statement.
While there is limited evidence to suggest that whale sharks are occasionally captured in gillnets the numbers captured per year or the frequency of these events are not presented or discussed (although it appears that efforts are underway to quantify this). Gillnets are usually highly size and shape selective and designed to mesh certain sized sharks/fish. The shape of a whale sharks head would make it virtually impossible to be meshed in a gill net and while there is the potential for entanglement the likelihood of this occurring is small. It is also not clear how much overlap there is between where the nets are set (what is the average net length) and the habitat whale sharks use. Data from satellite tracks could be used to determine the average distance offshore and therefore to determine overall overlap with gill nets. It would be good to see this attempted given that data are available. Furthermore this coastline is not protected and setting gillnets up to 500 m offshore in swell and over reef would be very difficult. I acknowledge that even low levels of fishing mortality could be unsustainable, however given that no data on the probability of capture are presented the link between population declines and gillnets appears over-stated. A more detailed explanation of how the gillnets are set (e.g. are they set from boats, are they set offshore in mid water, are they anchored or set as tangle nets, what mesh size is used and what water depth of water they are set in) is required to enable the reader to contextualise this information.

Overall, this manuscript is well written and the analysis are sound. The short track duration is problematic and at times the conclusions in relation to habitat use, diving and temperature profiles are over-stated given the limited track length as well issues with how much data are obtained from satellite tags.

---

## Round 0.2 · Minor Revisions

Both reviewers found the manuscript greatly improved. The only outstanding issues pertain to the gill net data as described by reviewer 2. Please address these issues, including clarifying the mesh size of the nets used in the present study. Also, the reviewer indicates that the scope of observations from this paper in terms of entanglements may not be representative of those from other areas. As such, this must be clarified.

·

Basic reporting

The paper is really clear and unambiguous. Background information provided is sound and sufficient, and the paper (including the abstract) is well structured.

Experimental design

The research is suitable for the scope of the journal. The research question is well defined and relevant, and the results fill an identified knowledge gap. The investigation is rigorous, methods have been greatly improved.

Validity of the findings

Data is robust and sound, and conclusions are well stated.

Additional comments

The paper has been clearly improved and the authors have addressed comments.

Reviewer 2 ·

Basic reporting

The authors have done a good job responding to the reviewer comments.

Experimental design

no comment

Validity of the findings

see comment below regarding gillnet mesh size and implications for whale shark entanglement

Additional comments

Overall the authors have done a good job addressing reviewer comments.

There is still a lack of understanding about gillnets and this needs to be addressed.

8-10 cm (3-4 inch) mesh is not large. This size gillnet is used to target small - medium sized fish (~30 – 50 cm TL) depending on the body shape. Nets this size are not even used to target small sharks and these nets do not have the capacity to regularly entangle sharks > 1 m TL. They certainly do not have the capacity to regularly entangle whale sharks unlike large mesh tangle nets (mesh size 20 – 40 cm) which have been implicated in whale shark capture in other parts of the world. I acknowledge that these nets may entangle the occasional whale shark and that this could have long term consequences but these nets simply will not results in frequent entanglement.

It is unclear whether observations but S. Pierce relate to mortality/entanglement along the Mozambique coastline or other parts of the world. It is also not clear if sharks were entangled in similar/same sized mesh nets. This needs to be made clear as entanglement and injuries reported by Speed et al. are not specific to Mozambique.

I suggest changing "large mesh" to small or medium size mesh to better reflect mesh size and the selectivity of different sized mesh.

---

## Round 0.3 · accepted · Accept

The authors have addressed the remaining issues and the manuscript is acceptable for publication in PeerJ.